# Co-Isolation and Characterization of Two Pandoraviruses and a Mimivirus from a Riverbank in Japan

**DOI:** 10.3390/v11121123

**Published:** 2019-12-04

**Authors:** Motohiro Akashi, Masaharu Takemura

**Affiliations:** Laboratory of Biology, Department of Liberal Arts, Faculty of Science, Tokyo University of Science, Shinjuku, Tokyo 162-8601, Japan

**Keywords:** pandoravirus, mimivirus, giant virus, DNA polymerase, intein, replication cycle, virus diversity

## Abstract

Giant viruses, like pandoraviruses and mimiviruses, have been discovered from diverse environments, and their broad global distribution has been established. Here, we report two new isolates of *Pandoravirus* spp. and one *Mimivirus* sp., named *Pandoravirus hades*, *Pandoravirus persephone*, and *Mimivirus* sp. isolate styx, co-isolated from riverbank soil in Japan. We obtained nearly complete sequences of the family B DNA polymerase gene (*polB*) of *P. hades* and *P. persephone;* the former carried two known intein regions, while the latter had only one. Phylogenetic analysis revealed that the two new pandoravirus isolates are closely related to *Pandoravirus dulcis*. Furthermore, random amplified polymorphic DNA analysis revealed that *P. hades* and *P. persephone* might harbor different genome structures. Based on phylogenetic analysis of the partial *polB* sequence, *Mimivirus* sp. isolate styx belongs to mimivirus lineage A. DNA staining suggested that the *Pandoravirus* spp. asynchronously replicates in amoeba cells while *Mimivirus* sp. replicates synchronously. We also observed that *P. persephone-* or *Mimivirus* sp. isolate styx-infected amoeba cytoplasm is extruded by the cells. To the best of our knowledge, we are the first to report the isolation of pandoraviruses in Asia. In addition, our results emphasize the importance of virus isolation from soil to reveal the ecology of giant viruses.

## 1. Introduction

Since the first report on *Acanthamoeba polyphaga mimivirus* (APMV), a member of the *Mimiviridae* family, in 2003, various families of giant viruses have been discovered around the world [1,2]. The first member of the *Marseilleviridae* family was found in a cooling tower in Paris in 2009 [3]. Tokyovirus, which belongs to the *Marseilleviridae* family, was the first giant virus isolated in Japan [4]. Both *Mimiviridae* and *Marseilleviridae* form icosahedral structures. The first family member of *Pithoviridae*, *Pithovirus sibericum*, found in 30,000-year-old Siberian permafrost, form amphora-shaped particles that are 1 µm in length; they are entirely different from other giant viruses in shape [5]. Recently, The Tupanvirus soda lake and deep ocean strains were isolated in Brazil from a soda lake and the Atlantic Ocean, respectively, and they form the sister group of *Mimivirudae* [6]. Tupanvirus virions form mimivirus-like icosahedrons surrounded by long fibers connected to a long cylindrical tail [6]. Medusavirus was recently isolated from hot spring water in Japan. This virus is thought to establish a new family *Medusaviridae,* although this has an icosahedral capsid similar to other giant viruses [7]. The continuous discovery of these giant viruses of various shapes suggests that they are quite diverse and may be ubiquitous.

The family *Pandoraviridae* is a group of giant viruses possessing the largest genomes and are found in various environments. In 2008, an endosymbiont of *Acanthamoeba* called ″KLaHel” was isolated from contact lenses and the fluid in the contact lens storage case of a patient with keratitis in Germany; no one knew this lifeform was a virus at that time [8]. KLaHel had an elliptical shape that was 1.0–1.2 µm in size, was surrounded by a 55–100 nm thick wall and had a prominent pore that was 55–60 nm in diameter [9]. In 2013, *Pandoravirus salinus* and *Pandoravirus dulcis*, which were isolated from marine sediment in a river mouth in Chile and mud from a pond in Australia, respectively, were reported to have 2.5-Mb, high GC-content (>60%) genomes [10]. After this discovery, KLaHel was revealed to be, in fact, a pandoravirus and was named *Pandoravirus inopinatum* [11,12,13]. In addition to these three viruses, many pandoraviruses have been found in various locations: *Pandoravirus brasiliensis*, *Pandoravirus pampulha*, *Pandoravirus massiliensis*, *Pandoravirus tropicalis*, and *Pandoravirus kadiweu* in Brazil; *Pandoravirus massiliensis* in New Caledonia; *Pandoravirus macleodensis* in Australia; and *Pandoravirus quercus* and *Pandoravirus celtis* in France [14,15,16,17,18]. Fifty percent of the known members of the *Pandoraviridae* family were isolated from soil or sediment samples [8,10,14,16,17,18]. As such, members of the *Pandoraviridae* family, as well as other giant viruses, are expected to have broad global distributions.

Pandoraviruses present an open pangenome with strain-specific genes that have no known homologs in other organisms or viruses [17]. These genomes have distinct sizes (1.8–2.5 Mb), but the genes are highly conserved [17]. Pandoravirus genes, both core genes and newly acquired/created genes, undergo negative selection pressure, suggesting that the newly acquired/created genes diversify the species of this family [18].

Here, we report the discovery of two *Pandoravirus* spp. and one *Mimivirus* sp., co-isolated from a small amount of soil from a riverbank in Japan. This is the first reported isolation of members of the family *Pandoraviridae* in an Asian country. We also report morphological, phylogenetic, and fluorescent microscopic analyses of these viruses.

## 2. Materials and Methods

### 2.1. Sample Collection and Virus Isolation

Soil sample a few centimeters from the surface was collected from the bank of Arakawa river (35°42′39.2″ N 139°50′54.7″ E). One spoonful of soil (less than 3 g) was suspended in 15 mL distilled water, and 4.5 mL of this sludge was mixed with 4.5 mL 2× PYG medium (4% Proteose peptone [*w/v*]; 0.2% Yeast extract [*w/v*]; 0.8 mM CaCl_2_ 8 mM MgSO_4_·7H_2_O; 5 mM Na_2_HPO_4_·12H_2_O; 5 mM KH_2_PO_4_; 0.2% Sodium citrate·2H_2_O [*w/v*]; 0.1 mM Fe(NH_4_)_2_(SO_4_)_2_·6H_2_O; 200 mM glucose; pH 6.5) with approximately 1.0 × 10^4^ cells of *Acanthamoeba castellanii* Neff (ATCC 30010^TM^), and antibiotics (1 mg/mL gentamicin, 10% penicillin/streptomycin, and 5% fungizone). This mixture was cultured in a 96-well plate and incubated at 26 °C; we then observed cytopathic effects (CPE) on the amoeba cells to monitor the infection progress. Virus samples were cloned by serial dilution and were proliferated in amoeba cells as described previously [19]. Virus-containing supernatants were initially collected from the amoeba-virus co-cultures after centrifugation (2000× *g*, 5 min). To collect the virus, the supernatant was further subjected to centrifugation (8000× *g*, 35 min) and diluted in distilled water. The viral titers were measured by determining the 50% tissue culture infective dose (TCID_50_) using a TCID_50_ calculator v.2.1 [20].

### 2.2. Sequencing Analysis of the Family B DNA Polymerase (polB)

*polB* sequences were confirmed by PCR and capillary sequencing. Primers used in this analysis are described in Appendix A. Viral genomic DNA was purified from viral particles using the NucleoSpin Tissue XS (Macherey-Nagel GmbH and Co. KG, Düren, Germany). PCR conditions were as follows: initial denaturation at 94 °C for 1 min; 35 cycles of denaturation at 98 °C for 10 s, annealing at 55 °C for 15 s, and extension at 68 °C for 30 s; and a final extension at 68 °C for 7 min. The reaction was performed using Tks Gflex DNA polymerase (TaKaRa Bio Inc., Shiga, Japan). Sequence analyses were performed by FASMAC Co. Ltd., Kanagawa, Japan. These sequences were assigned to the GenBank (Accession ID; LC511687, LC511688, LC511689).

### 2.3. Molecular Phylogenetic Analysis

For phylogenetic analysis of the *polB* nucleotide sequences of the *Pandoraviridae* family, we used the following sequences tagged by their NCBI accession IDs and the positions of the open reading frame on the nucleotides: *P. celtis* (MK174290, positions: 470867–473579/473730–475081); *P. dulcis* (NC_021858, positions: 565353–569634/569773–571127); *Pandoravirus inopinatum* (KP136319, positions: complement of 1758080–1760014/1760308–1760613/1760771–1764571/1764736– 1765050); *P. macleodensis* (MG011691, positions: 439503–444003/444093–446617); *Pandoravirus neocaledonia* (NC_037666, positions: 451521–456066/456184–458720); *P. quercus* (NC_037667, positions: 470524–473236/473387–474738), *P. salinus* (NC_022098, positions: 486737–491174/491313–493903), *Pandoravirus hades* (LC511687.1, 3–3749/3857–6284), *Pandoravirus persephone* (LC511688.1 3–2201/2339–4757).

Separated open reading frames (ORFs) were concatenated manually. Nucleotide sequences were initially translated to the amino acid sequences and aligned using the MAFFT program (v7.419) with ″--genafpair --maxiterate 1000--ep 0” options [21]. Thereafter, alignment data were back-translated to nucleotide sequences. Translation and back-translation were performed using the transeq program and the transalign program of EMBOSS tools (v6.6.0.0), respectively [22]. The overview of the aligned sequences was displayed using Jalview [23]. Subsequently, this alignment data was trimmed using pgtrimal program (v2.0.2016.09.06) in Phylogears tools [24] with ″—frame = 1 –method = strict” options, and the trimmed data were separated into five alignments based on the five assigned regions (region 1, positions: 1–1915; region 2: 1916–3283; region 3, 3284–3607; region 4, 3608–4609; region 5, 4610–5384). Finally, these five sets of data were subjected to a nucleotide composition test to confirm that there were no statistically significant differences in base composition among the sequences (*p* > 0.1) using the pgtestcomposition program in Phylogear tools. Phylogenetic analysis for each *polB* region was performed using the RAxML program (v8.2.9) for maximum likelihood trees and MrBayes program (v3.2.6) for Bayesian Markov chain Monte Carlo (MCMC) trees [25,26]. Evolutionary models for constructing trees were selected using the Kakusan4 program (v4.0.2016.11.07) [27]. The following models were selected: using RAxML: GTR_Gamma for all regions; using MrBayes: region 1/5: GTR_Gamma; region 2: HKY85_Invariant; region 3/4: HKY85_Gamma. In the maximum likelihood tree analyses, 1000 bootstrap replications were performed to estimate branch support. MCMC tree analyses were performed with the following conditions; 10,000,000 chains, burn-in = 1,001,000, and every 1000, and 40,002 trees were sampled for constructing posterior probability consensus trees. Phylogenetic trees were displayed using the Figtree software (v.1.4.4) [28].

Pairwise sequence identities (%) between sequences in the *polB* alignment were calculated using the Clustal Omega program [29]. For the calculation, the trimmed *polB* sequence alignment that was used for the phylogenetic analyses was used with the ″--full --percent-id” options. For clustering and heatmap visualization, the heatmap2 function of the gplots library in the R program (v3.5.0) was used [30].

For the phylogenetic analysis of mimiviruses *polB*, we performed the above-mentioned method while excluding the trimming process. NCBI database accession IDs of the aligned sequences of mimiviruses were as follows: *Acanthamoeba polyphaga mimivirus* (NC_014649.1, positions: 411696–412346); *Tupanvirus* soda-lake (KY523104.1, positions: 493719–493787); *Tupanvirus* deep ocean (MF405918.1, positions: 498404–498472); *Samba virus* (KF959826_2, positions: 411696–412346); *Mimivirus terra 2* (KF527228.1, positions: 175783–176433); and *Mimivirus* sp. isolate styx (LC511689.1, positions: 3–653). Inferred models for these analyses were as follows: RAxML: GTR_Gamma; MrBayes: HKY85_Homogeneous.

### 2.4. Random Amplified Polymorphic DNA (RAPD) Analysis

RAPD analysis was performed as previously described [31]. RAPD 10mer Kits A was obtained (Eurofins Genomics K.K., Tokyo, Japan). The six primers used in this analysis are listed in Appendix A (set 1: RAPD_1 and RAPD_2; set 2: RAPD_1 and RAPD_3; set 3: RAPD_1 and RAPD_4; set 4: RAPD_1 and RAPD_5; set 5: RAPD_1 and RAPD_6). PCR conditions were as follows: initial denaturation at 94 °C for 1 min; 35 cycles of denaturation at 98 °C for 10 s, annealing at 30 °C for 15 s, and extension at 68 °C for 30 s; final denaturation at 68 °C for 7 min. The reaction was performed using the Tks Gflex DNA polymerase (TaKaRa Bio, Kusatsu, Shiga, Japan).

### 2.5. Transmission Electron Microscopic Observation

Cell fixation was performed as described previously [4]. Amoebae were inoculated with viruses at an M.O.I. of 1.0 and incubated at 26 °C. These samples were collected at 10 and 24 h.p.i. Cells were washed with ice-cold PBS twice, fixed with ice-cold 2% glutaraldehyde solution three times, and then stored at 4 °C overnight. Fixed cells were washed with PBS three times and stained using ice-cold 2% osmium tetroxide for 1 h. The stained cells were dehydrated with increasing concentrations of ethanol (50%, 70%, 80%, 95%, 100%) for 5 min and with propylene-oxide for 10 min at room temperature. Dehydrated cells were embedded in Epon 812 (TAAB Laboratory Equipment, Berks, UK), and resin curing was done at 60 °C for two days. Ultra-thin sections were analyzed using a transmission electron microscope (JEM-1400; JEOL, Tokyo, Japan, or H-7600; Hitachi, Tokyo, Japan) at the Hanaichi UltraStructure Research Institute (HUSRI).

### 2.6. DNA Staining

*A. castellanii* cells were infected with the viruses at an M.O.I. of 10 in PYG medium on 24-well plates and incubated at 26 °C. M.O.I was calculated using the following formula: [M.O.I] = [Plaque forming units (pfu)]/[number of cells]. The value of the plaque-forming unit (pfu/mL) was approximated by the value of TCID50(/mL). Samples were collected at 4, 6, 8, 10, and 24 h. p. i. Cells were fixed with 2% glutaraldehyde at 4 °C for 10 min, stained with 500 ng mL^−1^ of 4′6-diamidino-2-phenylindole (DAPI) solution for 1 min. Fluorescence microscopic analysis was performed using BX50 microscope (Olympus, Tokyo, Japan). The brightness/contrast of compared images was identically adjusted and merged using the Fiji software [32]. The settings were: light: 8-bit color, brightness/contrast minimum and maximum: 10–150; DAPI: 8-bit color, brightness/contrast minimum and maximum: 10–150.

### 2.7. Time-Lapse Analysis of Mimivirus-Infected Amoeba Cell

*Mimivirus shirakomae* [33] was co-cultured with *A. castellanii* in PYG medium in a 25-cm^2^ flask for 24 h. Infected amoeba cells were collected through centrifugal separation (1200 rpm, 5 min) and resuspended in PYG medium. This culture was inoculated into a 96-well plate, and the CPE cells were observed using an inverted phase-contrast microscope (Eclipse TS100, Nikon Corporation, Tokyo, Japan). An image was automatically taken every minute for 14 h. These pictures were displayed for 0.0125 s in the movie (4800× speed).

## 3. Results

### 3.1. Two Pandoraviruses and One Mimivirus Co-Isolated from a Riverbank Soil Sample

We isolated two pandoraviruses and one mimivirus from less than three grams of soil from the bank of the Arakawa river in Japan. Virus candidates were proliferated using *A. castellanii* as host and were later purified. The isolates were identified as *Pandoravirus* spp. and *Mimivirus* sp. by polymerase chain reaction (PCR)-based sequence analysis using virus-specific primers.

We named the two soil-derived pandoravirus isolates as *P. hades* and *P. persephone*, after the king and queen of the underworld in Greek mythology. We also tentatively named the mimivirus as *Mimivirus* sp. isolate styx; the name ″styx” is derived from the goddess of the river that flows through the underworld in Greek mythology. This was the first report on the isolation of a *Pandoraviridae* family virus from an Asian country.

Transmission electron microscopic (TEM) analysis demonstrated that the three isolates were typical pandoraviruses and mimivirus. As expected, *P. hades* and *P. persephone* were almost 1 µm in length, with round-oval shape and an apex-like aperture (Figure 1a,b). The *Mimivirus* sp. isolate styx capsid had a diameter of almost 500 nm and was surrounded by long outer fibers as expected (Figure 1c). The nucleocapsid packaging process was different between the pandoravirus and the mimivirus, as described previously [10,15,34]. For the mimivirus, the capsid was already filled with viral components during the viral maturation process (Figure 1f). However, for the pandoraviruses, we observed a gap between the immature capsids and nucleocapsids (Figure 1d,e). These empty spaces were quite similar to the space that has been previously shown in an immature *P. salinus* particle [10]. It has been reported that the capsid assembly of pandoraviruses also starts at the end of the particle that is opposite the ostiole-like apex [15]. We also observed this type of morphogenesis from *P. hades* and *P. persephone* with the vacant space between the immature capsid and nucleocapsid at the non-apical end of the particles (Appendix A). As such, through TEM analysis, two isolates were visually identified as members of the *Pandoraviridae* family, and one isolate was a member of the *Mimiviridae* family.

The pandoravirus early virus factories (VFs) were light-colored (electron-lucent) regions in the cytoplasm (Figure 2a,b). Electron-lucent VFs have been observed in other pandoraviruses [15]; thus, this characteristic is shared among members of the *Pandoraviridae* family. In contrast, the VF of the mimivirus was easily identifiable (Figure 2c), as it had a higher electron density than the rest of the amoeba cytoplasm. Furthermore, similar to a previous report, the mature VFs and the virus particles of our pandoraviruses homogenized into the cytoplasm; there were no identifiable boundaries between the cytoplasm and the VFs (Figure 2d,e) [14]. Contrastingly, mimivirus-infected cells had compact VFs surrounded by several virus particles in late stages of infection (Figure 2d,f).

### 3.2. Sub-Classification of Isolated Viruses

To elucidate the phylogenetic relationships of *P. hades* and *P. persephone* with known pandoraviruses, we performed phylogenetic analyses based on the family B DNA polymerase gene (*polB*) of these viruses. First, we obtained the sequence of *polB* in our isolates. Lengths of the *polB* partial sequences of *P. hades* and *P. persephone* were 6284 bp and 4757 bp, respectively. Generally, the *polB* gene of pandoraviruses includes two ORFs [10]. The *polB* of the two isolates encoded two ORFs with a short spacer region; the spacer length of *P. hades* and *P. persephone* were 107 bp and 137 bp, respectively. The *polB* of *P. salinus* has been reported to harbor two intein sequences [10]; we temporarily named these as intein 1 and intein 2 (Figure 3a). *polB* of *P. hades* also harbored both inteins, while that of *P. persephone* only contained intein 2 (Figure 3a). For comparison, with regard to the absence of intein 1, *P. celtis* and *P. quercus* also do not harbor intein 1, as in *P. persephone* (Figure 3a).

Phylogenetic analysis of *polB* in pandoraviruses revealed that the two isolates were phylogenetically close to *P. dulcis*. We separately analyzed five *polB* regions (corresponding to intein 1 and 2, and the three regions interrupted by the inteins), which we tentatively named as regions 1–5 (Figure 3a). We analyzed the *polB* separately as we speculated that the inteins might have co-evolved with the PolB catalytic regions and might have affected them. We tested two methods for reconstructing the phylogenetic trees: the maximum likelihood and the Bayesian method. We obtained the same topology trees from both methods, with significant support for almost all branches (Figure 3b and Appendix A). The *Pandoraviridae* family can be classified into clades A and B based on the *polB* gene. Clade A includes *P. quercus*, *P. inopinatum*, *P. salinus*, and *P. dulcis*, and clade B includes *P. neocaledonia* and *P. macleodensis* [17]. Phylogenetic trees reconstructed from regions 1 and 5 showed almost the same topology as expected. These results indicate that both *P. hades* and *P. persephone* are members of clade A (Figure 3b), and they are most closely related to *P. dulcis*. Similar to the results of the phylogenetic analysis for regions 1 and 5, the tree for region 2 (intein 1) showed that *P. hades* is closely related to *P. dulcis*. Among the members of the *Pandoraviridae* family, *P. persephone* was the only virus that did not have region 2 (intein 1) and only had the conserved region 4 (intein 2), which was closely related with that of *P. hades* (Figure 3a). The tree derived from region 3 exhibited a different topology from those derived from regions 1 and 5. The group without region 4 (intein 2) formed a group (clade A) on the region 3-based tree, which included *P. dulcis*, *P. quercus*, and *P. celtis*, indicating that region 3 correlated with the presence or absence of intein 2 (Figure 3b). As such, *P. hades* and *P. persephone* possess different *polB* structures.

Furthermore, comparing the pairwise identities of *polB* sequences showed the detailed relationships between the new isolates and pandoraviruses for which sequence information is already available. The pairwise identity between *P. hades* and *P. dulcis* was 98.5%, indicating that these viruses are not identical but are closely related (Appendix A). In contrast, *P. persephone* and *P. dulcis* exhibited the lowest identity value (72.1%, Appendix A), even though they are phylogenetically close (Figure 3b). The value between *P. hades* and *P. persephone* was 96.7% (Appendix A), which is lower than that between *P. hades* and *P. dulcis*. Additionally, the percent identity between *P. persephone* and *P. macleodensis*, which belongs to pandoravirus clade B, was slightly higher than between *P. persephone* and the other clade A pandoraviruses (84.4%, Appendix A), suggesting that *P. persephone* is a clade A virus with a highly conserved *polB* sequence that is more similar to the ancestral sequence of both clade A and clade B pandoraviruses.

Additionally, we noted that *P. hades* and *P. persephone* harbor different genome structures. Random amplified polymorphic DNA (RAPD) analysis is a suitable method to evaluate genetic diversity among subspecies [35], so we applied this method to compare the two pandoravirus genomes. As a result, we observed different banding patterns between the two pandoraviruses for every tested primer set (Figure 4). Based on these results, *P. hades* and *P. persephone* have distinct genome structures.

We also sequenced a part of the *polB* sequence of *Mimivirus* sp. isolate styx. Phylogenetic analysis of this sequence with the *polB* of other known mimiviruses revealed that *Mimivirus* sp. isolate styx belongs to lineage A because its intein-*polB*-concatenated region was identical to that of APMV (Appendix A). Taken together, based on phylogenetic analysis, it is clear that we isolated two *Pandoravirus* spp. and a *Mimivirus* sp. from one soil sample. Whole-genome sequence analysis of these viruses in future studies could provide more details about their evolutionary histories.

### 3.3. Isolated Pandoraviruses Proliferate Asynchronously in Amoeba Cells

Virus factor formation in the host is an index of giant virus infection and its replication, and these patterns slightly differ between virus species. We infected amoeba cells with *P. hades*, *P. persephone*, or *Mimivirus* sp. isolate styx at a multiplicity of infection (M.O.I.) of 10.0 and stained the infected cells with 4′,6-diamidino-2-phenylindole (DAPI) at 4, 6, 8, 10, and 24 h post-infection (h.p.i.) in order to identify the stage of infection by nucleotide content (Figure 5). Unmerged photos are shown in Appendix A. We found that for the pandoraviruses, the well-stained cells coexisted with partially stained ones in samples from all time points (Figure 5 and Appendix A), while the *Mimivirus* sp. isolate styx VFs formed almost at the same time point 6 h.p.i. These results suggest that pandoravirus replication occurred asynchronously in amoeba cells (i.e., one pandoravirus-infected cell has a high degree of viral replication while another does not) as opposed to the synchronous infection observed for *Mimivirus* sp. isolate styx. Therefore, pandoraviruses and mimivirus differ in the synchronization of infection and proliferation in amoebas.

We occasionally observed that some *Mimivirus* sp. isolate styx-infected cells ejected the VFs (Figure 6), causing free VFs in the culture media, particularly at 24 h.p.i. (Figure 5). A similar situation was observed in *P. persephone*-infected cells, but these cells seemed to have exhausted cytoplasm because VFs filled the cells (Figure 6). We observed that the putative VF of *Mimivirus shirakomae* was spouted out when the infected cell had lysed (Appendix A), and this particle might resemble the VFs shown in Figure 5 and Figure 6. Therefore, the VFs of mimiviruses probably usually leave the infected amoeba cells from the mimivirus-infected amoeba cells.

## 4. Discussion

Here, we report three novel giant viruses (two pandoraviruses and one mimivirus) that were co-isolated from a soil sample from a riverbank in Japan. Considering the fact that these viruses were isolated from less than 3 g of soil, these viruses presumably co-inhabit the soil environment. This is not the first isolation of pandoravirus from soil; *Pandoravirus pampulha*, *P. massiliensis*, *P. quercus*, and *P. celtis* have been isolated from soil [14,16,17,18]. Thus, soil is presumably a habitat of one of the natural hosts of pandoraviruses. Results of our *polB* sequence analysis and RAPD analysis (Figure 3 and Figure 4) suggest that the pandoravirus genome is quite variable. Pandoraviruses have distinctive genome sizes (1.8–2.5 Mb), although the position of orthologous genes on their genomes are similar.

Phylogenetic analyses of *polB* sequences were used to estimate the differentiation processes of pandoraviruses. The topology of the phylogenetic tree of *polB* region 3, which was conserved in all species analyzed in this study, did not correspond to other *polB* coding regions (region 1/5) but correlated to the presence or absence of intein 2 (region 4). This suggests that region 3 is highly affected by intein 2 and that the deletion of intein 2 causes the mutation or speciation of the region 3 sequence (Figure 3b). In other words, region 3 possibly relates to the mobility of intein 2. Interestingly, *P. persephone* possesses intein 2 but does not harbor intein 1 (region 2, Figure 3a). Although *P. celtis* and *P. quercus* also do not have intein 1, they are not phylogenetically close to *P. persephone*. Thus, we assume that the deletion of intein 1 in *P. persephone* independently occurred from those in *P. celtis* and *P. querqus* (Figure 3a,b). The sequence of the lost intein 1 in *P. persephone* was probably similar to the sequence of intein 1 in *P. hades* and *P. dulcis* because *P. persephone, P. hades*, and *P. dulcis* are phylogenetically close each other. In addition, based on the pairwise identities of the *polB* sequences, *P. hades* is more similar to *P. dulcis* than to *P. persephone* (Appendix A), which supports the phylogenies derived from *polB* region 1 and region 5 (Figure 3b). The *polB* gene of clade A and B pandoraviruses have been suggested to have undergone purifying selection, and the dN/dS value of clade A has been reported to be smaller than that of clade B [17]. In general, inteins are known as selfish genetic elements and have little influence on their target gene fitness. However, the phylogenetic analysis of *polB* showed that the sequence of region 3 corresponds to the presence/absence of intein 2 (Figure 3b). This suggests that the intein 2 affects the gene fitness of *polB* gene. Although the effects of intein presence or absence on *polB* gene proliferation of pandoraviruses remain unclear, the clade A species seem to have lost their inteins. Therefore, clade A pandoraviruses seem to be evolving through the loss of inteins in the *polB* gene. 

DAPI staining showed that, even at high titers of infection, DNA replication of pandoraviruses was asynchronously activated in the host cells in contrast to that of the mimivirus (Figure 5). According to a previous report, the cytoplasm of pandoravirus-infected cells was filled with VFs [15]. We also observed pandoravirus-filled amoeba cytoplasm in *P. hades*- or *P. persephone*-infected cells after TEM analysis and after DAPI staining (Figure 2 and Figure 5). The cells in which the pandoravirus VFs have spread throughout the cytoplasm had strong DAPI signals, suggesting that DNA replication of pandoravirus was activated during the VF enlargement process (Figure 5). On the other hand, VFs of mimivirus contained high-density DNA, which were tightly compacted and surrounded by immature virus particles, suggesting that VFs in mimivirus pack their own DNA replication machineries (Figure 2f, Figure 5 and Appendix A). These differences observed between the mimivirus and the pandoraviruses may be due to differences in the DNA replication factors encoded in the virus genomes. Mimiviruses may carry sufficient factors for DNA replication, while pandoraviruses may not and may instead depend on host replication factors.

Our study followed a novel methodology for the analysis of pandoraviruses and other giant viruses, which could be used as a guideline for the exploration of new giant viruses in the future. We revealed that region 1 and the region 5 of *polB* gene are genetically stable (Figure 3), and can be used as a marker for pandoravirus. These results are important in the field of microbial ecology, in particular for the construction of primer sets for meta-barcoding analyses. Although RAPD is an old technique, our results demonstrate that it is still effective. More giant viruses will be isolated in the future, and the field of giant virus ecology will expand further. Under these circumstances it would be necessary to use techniques such as whole-genome sequence analysis for further characterization of the virus species. A minimalistic and fast approach like RAPD would be effective in these scenarios.

Taken together, in this study, we isolated two novel pandoraviruses and one novel mimivirus from soil taken from a riverbank in Japan. To our knowledge, we are the first to report the isolation of pandoraviruses in an Asian country. The phylogenetic characterizations of these isolates revealed their relationships with other known pandoraviruses and mimiviruses. Our findings suggest that giant viruses, particularly pandoraviruses, likely thrive in widely different environments by mutating their genes.

## Figures and Tables

**Figure 1 viruses-11-01123-f001:**
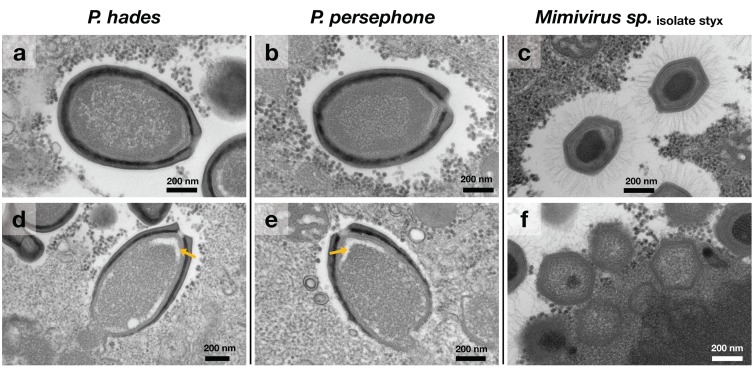
Virus particles of *Pandoravirus hades*, *P. persephone,* and *Mimivirus* sp. isolate styx observed by transmission electron microscopy. (**a–c**) Mature virus particles; (**d–f**) virus particles in the maturing processes of *P. hades*, *P. persephone*, and *Mimivirus* sp. isolate styx, respectively. Scales are shown in each figure. Yellow arrow: vacant space between the immature capsid and nucleocapsid.

**Figure 2 viruses-11-01123-f002:**
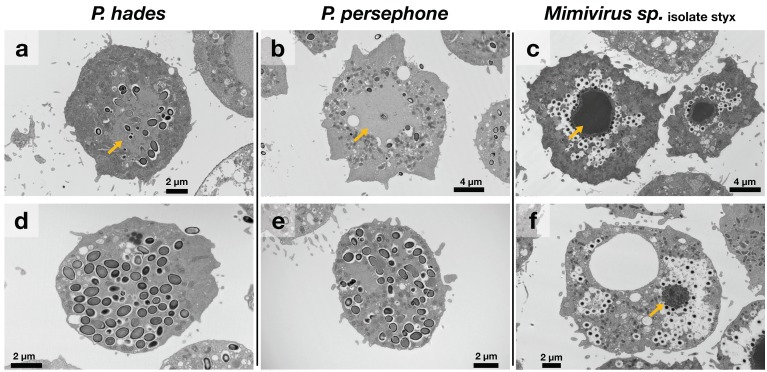
Early and late stages of virus factories (VFs) of *P. hades*, *P. persephone,* and *Mimivirus* sp. isolate styx observed using transmission electron microscopy. (**a–c**) Early VFs and (**d–f**) mature VFs of *P. hades*, *P. persephone*, and *Mimivirus* sp. isolate styx, respectively. Yellow arrow: VFs. Mature VFs of *P. hades*, *P. persephone* homogenized with the cytoplasm; therefore, VFs could not be differentiated.

**Figure 3 viruses-11-01123-f003:**
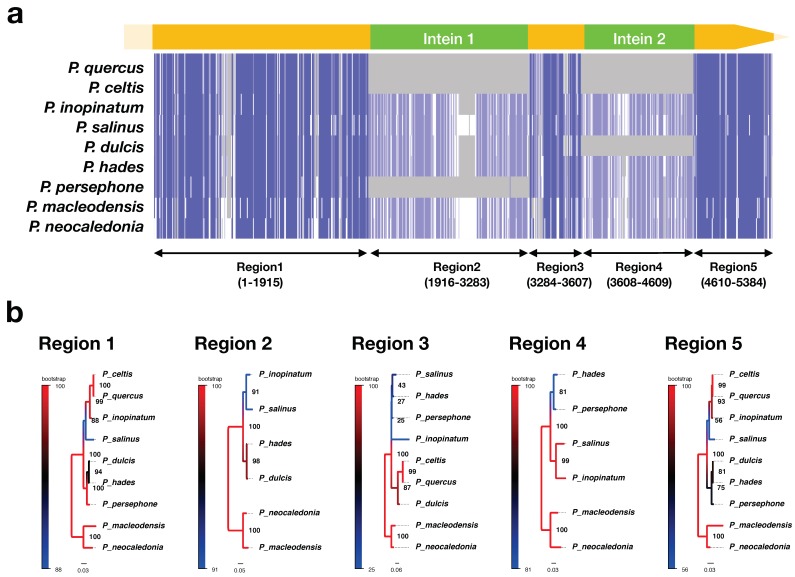
Sequence alignment and phylogenetic analysis of the *polB* gene of the *Pandoraviridae* family. (**a**) Overview of the nucleotide sequence alignment of the *polB* sequences. Top: yellow bars: *polB* regions, green bars: inteins, Middle: color scale from blue to white: high to low sequence identities of each site. Bottom: sequence regions of alignment corresponding to the phylogenetic trees shown below; (**b**) Maximum likelihood phylogenetic trees of the specific *polB* regions. Color scale and node number: bootstrap value. Scale bar: number of substitutions per site.

**Figure 4 viruses-11-01123-f004:**
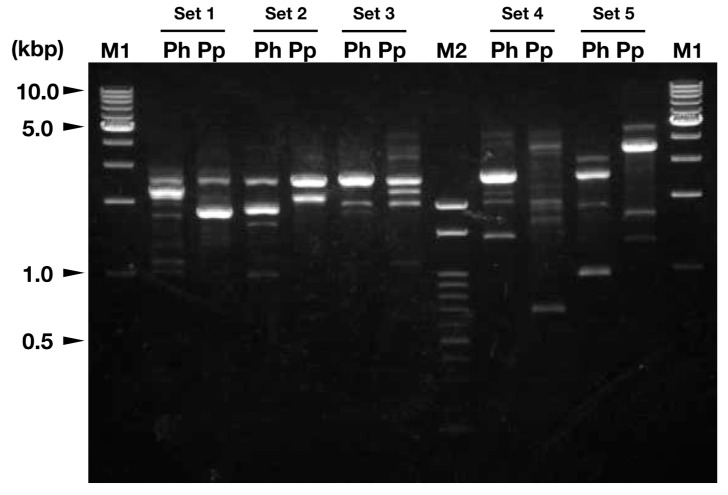
Random amplified polymorphic DNA (RAPD) analysis of *P. hades* and *P. persephone*. Set 1 to 5: Specific primer sets for each PCR reaction. Ph: *P. hades*. Pp: *P. persephone*. M1: OneSTEP Ladder 1kb (1–10 kbp; Nippon Gene Co., Ltd., Tokyo, Japan) M2: Gene Ladder 100 (0.1–2 kbp; Nippon gene Co., Ltd.).

**Figure 5 viruses-11-01123-f005:**
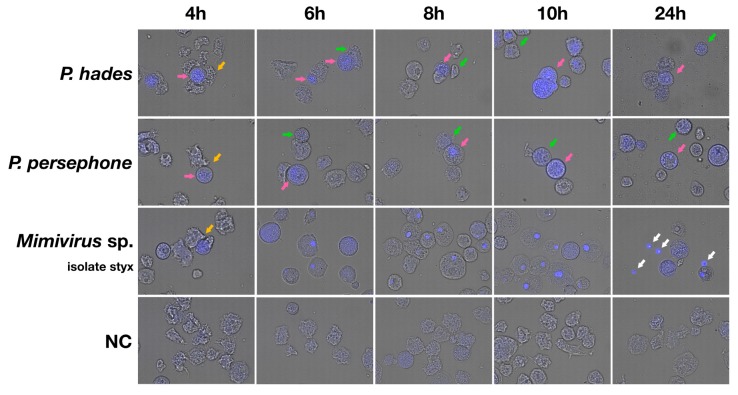
DNA staining analysis of *P. hades*, *P. persephone,* and *Mimivirus* sp. isolate styx. The time indicated (4, 6, 8, 10, and 24 h) corresponds to sample collection time points post-infection. White arrows: free virus factories. Yellow arrows: amoeba(s) adhering to cells that exhibit cytopathic effects. Green arrows: examples of partially stained cells. Pink arrows: examples of well-stained cells. NC: Uninfected amoeba cells as a negative control. Objective lens: ×40.

**Figure 6 viruses-11-01123-f006:**
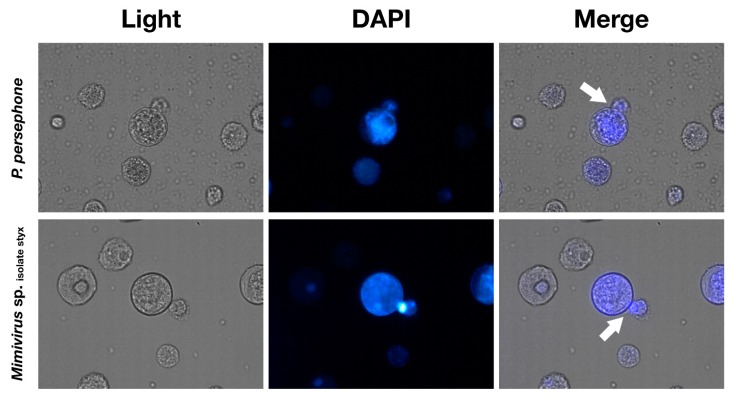
*Acanthamoeba castellanii* that exhibit cytopathic effects spout internal structures at 24 h.p.i (cytoplasm, VF, and virus particles). Infected viruses: *P. persephone* (above) and *Mimivirus* sp. isolate styx (below). M.O.I: 10. White arrows: spouted virus factory (VF) or cytoplasm. Objective lens: ×40.

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
