# Peer review of "Co-Isolation and Characterization of Two Pandoraviruses and a Mimivirus from a Riverbank in Japan"

_viruses, 2019, doi:10.3390/v11121123_

Round 1

Reviewer 1 Report

The study by Akashi and Takemura describes the successful co-cultivation of three giant viruses from a soil sample from a riverbank in Japan. The authors sequenced the polB genes of the new viruses and provide a phylogenetic classification; all three viruses are closely related to previously isolated giant viruses, two are pandoraviruses and one is a mimivirus. Furthermore, the authors investigated the infection process of these viruses using electron microscopy, light microscopy and DNA-stain based fluorecence microscopy. They also provide a short video showing lysis of a amoebal host by the newly isolated mimivirus. Overall, it is a solid study with sound methods which further expands the known global distribution of giant viruses. I support publication of this manuscript in Viruses.

Here some minor issues which need to be addressed prior to publication:

Life cell imaging. There is no video for pandoravirus infection and host lysis. It would be a great addition to the manuscript to have a video for pandoravirus, and, if possible, to use DAPI staining in both videos.

Why do the authors capitalize certain virus names; e.g. "two Pandoraviruses and a mimivirus" and not "two pandoraviruses and a mimivirus", or throughout the text "Pandoravirus" and "mimivirus".? L32-33: "Both the megavirus and the Marseillevirus", why Marseillevirus capitalized?

L14: "co-isolated river soil", "co-isolated from river soil"

L306-310: The authors write that uninfected amoeba (control) attach to each other, and that 4 hours post infection, vegetative and cytopathic amoeba attach as well.The significance of this observation is not perfectly clear.

L375-377: Are differences in genetic structure the reason that these viruses thrive in different environments, or can it simply be explained by the host range and thus the presence or absence of suitable hosts?

Figure S4: Only 5 reference sequences of Mimiviruses were included, many more are available. Additional taxon sampling might help to better resolve split between A. polyphaga mimivirus and mimivirus sp. isolate styx (this study).

Video S1: "immediately after blasting of the A. castellanii cell" "immediately after lysis of the A. castellanii cell"

Reviewer 2 Report

The results are not novel enough to deserve publication as original research. The three characterizations are much below today's standards for reporting new virus strains (see for instance Genome announcements), and even less to perform a phylogenetic analysis.

I would advise the authors to perform a complete (draft) sequencing of the genome of their three isolates, and only report new findings instead of confirming what was previously reported in numerous previous publications.

Reviewer 3 Report

This paper describes the discovery and characterization of three new giant viruses, two pandoraviruses and a mimivirus that is very similar to the first described mimivirus, APMV.  All of this from a mere teaspoon of soil at the edge of a river in Japan. These characterizations add to the growing body of literature on the emerging field of giant virus biology, and the ubiquity of giant viruses in soils and water.

The authors do an excellent job of describing their studies (though they need to define their use of “M.O.I.”). And, the data for the pandoraviruses are described sufficiently in the results such that it is clear that the viruses isolated are novel and have unique characteristics. They point out that the mimivirus isolated is very similar to APMV, which is a bit surprising given the significant differences in sample origins. The use of the polB gene as a distinguishing marker is insightful and may help to open up the discovery of pandoraviruses yet further, especially with the ability to recognize the 5-regions of the protein, somewhat delimited by the two inteins of certain isolates. This sequencing method (not really discussed) combined with an older but still effective technique of using RAPD analyses across the 5 regions of the polB protein clearly demonstrates the uniqueness of these isolates.

One unfortunate feature of the manuscript is the addition of the video in the supplementary materials. The virus used for this study is not part of the current study and should not be conflated with their good work presented. This type of video would be acceptable and informative if they had used isolated mimivirus characterized here, but otherwise should be deleted from the manuscript. Perhaps the authors would consider a separate publication for those studies.

Author Response

Response to Reviewer 3 Comments
Comments and Suggestions for Authors
Comments and Suggestions for Authors This paper describes the discovery and characterization of three new giant viruses, two pandoraviruses and a mimivirus that is very similar to the first described mimivirus, APMV. All of this from a mere teaspoon of soil at the edge of a river in Japan. These characterizations add to the growing body of literature on the emerging field of giant virus biology, and the ubiquity of giant viruses in soils and water.

The authors do an excellent job of describing their studies (though they need to define their use of “M.O.I.”). And, the data for the pandoraviruses are described sufficiently in the results such that it is clear that the viruses isolated are novel and have unique characteristics. They point out that the mimivirus isolated is very similar to APMV, which is a bit surprising given the significant differences in sample origins. The use of the polB gene as a distinguishing marker is insightful and may help to open up the discovery of pandoraviruses yet further, especially with the ability to recognize the 5-regions of the protein, somewhat delimited by the two inteins of certain isolates. This sequencing method (not really discussed) combined with an older but still effective technique of using RAPD analyses across the 5 regions of the polB protein clearly demonstrates the uniqueness of these isolates.
Response: Thank you for your helpful comments and suggestions. We have revised the manuscript based on your suggestions. The revised/newly inserted sentences are highlighted in yellow.

#line163 “M.O.I was calculated with following formula: [M.O.I] = [Plaque forming units (pfu)]/[number of cells]. The value of plaque forming unit (pfu/ml) was approximated by the value of TCID50(/ml).”

#line 372 “Our study followed a novel methodology for the analysis of pandoraviruses and other giant viruses, which could be used as a guideline for the exploration of new giant viruses in the future. We revealed that region 1 and the region 5 of polB gene are genetically stable (Figure 3), and can be used as marker for pandoravirus. These results are important in the field of microbial ecology, in particular for the construction of primer sets for meta-barcoding analyses. Although RAPD is an old technique, our results demonstrate that it is still effective. More giant viruses will be isolated in the future, and the field of giant virus ecology will expand further. Under these circumstances it would be necessary to use techniques such as whole-genome sequence analysis for further characterization of the virus species. A minimalistic and fast approach like RAPD would be effective in these scenarios.”

One unfortunate feature of the manuscript is the addition of the video in the supplementary materials. The virus used for this study is not part of the current study and should not be conflated with their good work presented. This type of video would be acceptable and informative if they had used isolated mimivirus characterized here, but otherwise should be
deleted from the manuscript. Perhaps the authors would consider a separate publication for those studies.
Response: Thank you for your comment. However, reviewer 2 claimed that this result is the only new information in this study. Therefore, we would like to ask you to allow us to keep this result.

Round 2

Reviewer 2 Report

Changes have only been made  in the text. In absence of more scientific substance in this work, I unfortunately see no reason to modify my previous judgement. I continue to think that the manuscript does reach the threshold of information content required for a scientific article in a reputable journal.